# The Adoption of Green Market Orientation in Logistic Industries—Empirical Evidence from Vietnamese SMEs

Quang-Huy Ngo

Business Department, FPT Greenwich Center, Can Tho Campus, FPT University, Can Tho City 900000, Vietnam; huynq121@fe.edu.vn

**Abstract:** Recently, the Vietnamese government established an ambitious plan that strongly promotes the development of the logistic industry. However, there is an environmental concern regarding this development. In addition, adopting environmental management practices may reduce the performance of small and medium enterprises (SMEs). Therefore, this study draws upon the natural resource-based and ambidextrous views of green innovation to examine the impact of green market orientation on performance. Particularly, this study proposes that adopting green market orientation permits logistic small and medium enterprises (SMEs) to improve exploitational and exploratory green innovation and, as such, results in the implications of organizational performance based on the balanced scorecard approach. Data were collected from 338 SMEs operating in the logistics industry. Partial least squares structural equation modeling was used to assess the data. The results indicate that the adoption of GMO permits these SMEs to indirectly improve their operational performance through exploitational and exploratory green innovation. These results provide practical implications for managers of logistic SMEs in Vietnam. In addition, this study also contributes to the literature by addressing the gaps in the existing literature.

**Keywords:** ambidextrous green innovation; green market orientation; logistic providers; organizational performance; SME; Vietnam

## 1. Introduction

Vietnam is located in Southeast Asia, one of the world's most dynamic regions. Before the COVID-19 pandemic, the Vietnamese economy enjoyed an average growth rate of more than 6.5% annually over 20 years [1]. Vietnam has one of the most integrating economies in the world, with over 170% of trade contributing to its GDP [2]. Vietnam is emerging as a global industrial production hub and a destination that strongly attracts foreign direct investment [3].

The growth of the Vietnamese economy is suggested to have a strong association with trade expansion. Its annual export growth is about 18%, which is triple its GDP growth [4]. In 2019, the Vietnamese export revenue reached USD 264 billion, which allowed Vietnam to become one of the leading exporting countries globally [5]. As a result, the Vietnamese government has recently set an ambitious goal of promoting export-related achievement to a higher degree. One of the key elements of this plan is to develop the logistics industry in Vietnam. For instance, the Vietnamese government passed Decree no. 200/QD-TTG, which aims to increase the growth rate of logistics services by 15–20% and the contribution of this sector to the GDP by 8–10% [6].

This development of a logistics service, however, may cause environmental concerns. Logistic activities are acknowledged as the causes of severe negative impacts on the environment, particularly freight transportation, a key activity in logistic sectors [7]. According to Doherty and Hoyle [8], it was estimated that logistic industries account for about 5.5% of greenhouse gases, and road freight is the major source. In Vietnam, the transportation of goods is mainly based on roads, which accounts for more than 95% of the greenhouse

gases emitted from total freight transportation [9]. In addition, the Vietnamese government has declared ambitious commitments to climate change at COP26 [10]. It aims to achieve net zero greenhouses gas emissions by the end of 2050 [11]. In this regard, the government is more likely to release environmental regulations to achieve such commitments.

In the environmental literature, it is suggested that two forces drive logistic providers' propensity to prioritize adopting environmental management practices to reduce negative impacts on the environment. First, the pressure from governmental regulation plays a crucial role in shaping these providers' inclination to adopt environmental management practices [12]. Second, the increasing concern about greenhouse gas emissions positively influences logistic providers' adoption of said practices [13]. In this regard, in Vietnam, the current scenario positively affects the degree to which logistic providers adopt environmental management practices.

One environmental management practice is green market orientation (GMO). It is suggested that organizations should align their operations in line with GMOs to address environmental issues such as ecological imbalance and depleting natural resources [14–16]. GMO adoption improves corporate greening, and as a result, it promotes the delivery of environmentally friendly products and services, which is valuable to customers [17]. GMO aids organizations via the extent to which it enhances their unique capabilities to achieve environmental goals [18].

GMO is a sustainability form of market orientation (MO). Borazon et al. [19] and Wang [18] borrow the MO concepts from Narver and Slater [20] to define GMO. It drives organizations to adopt green behaviors to create superior values for customers, and as a result, it allows for superior performance. The performance implications of this orientation are interpreted based on a resource-based view (RBV), as this orientation is a rare, valuable, and has inimitable resource-permitting performance implications. Recently, some authors drew upon the natural resource-based view (NBRV) to address the performance implication effects of GMO.

However, due to the novelty of this concept, there is a limited understanding of whether GMO impacts performance, particularly in logistics-focused small and medium enterprises (SMEs). The existing literature shows that the relationship between SMEs' adoption of environmental management practices and performance in terms of organizational effectiveness is mixed [21]. In addition, the literature suggests that reducing greenhouse gas emissions has contracting impacts on performance [22–24]. To the best of the authors' knowledge, no study has examined whether logistics SMEs that adopt GMO achieve OP. Therefore, this lacuna motivates the current study's exploration of this effect.

MO literature suggests that MO plays a crucial role in enhancing innovation. In GMO studies, MO has been revealed to have a positive impact on GI [18,19,25]. The ambidextrous view of green innovation [26] classified GI into two approaches: exploitational and exploratory GI [26–28]. However, to the best of the author's knowledge, no study addresses whether GMOs induce exploitational and exploratory GI. In this regard, the effects of GMO on both types of GI are underexplored. This presents another lacuna that has motivated the current study, which also assesses whether GMOs induce exploitational and exploratory GI.

In the MO literature, scholars have intensively examined the paradigm of MO innovation performance [29–36]. Examining the mediating effects allows for an insight into the mechanisms that enable MO's ability to enhance performance through innovation [37]. In addition, it is argued that the ambidextrous view sheds light on how MO fuels various types of innovation and, as such, impacts performance [36]. GMO is a novel concept. The extant literature shows a limited understanding of the paradigm of GMO-exploratory and exploitation-related GI-performance. Thus, this study questions whether exploitational and exploratory GI mediate the relationship between GMO and OP.

Therefore, this study aims to draw upon the natural resource-based and ambidextrous views of GI to examine the mediating effects of exploitational and exploratory GI on the relationship between GMO and OP. Accordingly, this study collected data by using the

survey method. Particularly, logistic SMEs operating in Vietnam are within the scope of this study. The partial least squares structural equation modeling technique was used to assess the data. The results indicate that two types of GI fully mediate the relationship between GMO and OP.

This study contributes to the literature in the following ways. First, this study makes a theoretical contribution by using NBRV as a theoretical lens to explain the mechanism by which exploitational and exploratory GI fully mediate the relationship between GMO and OP. In addition, by employing this view, this study also fills the gaps in the GI literature by indicating the antecedents and effects of GI. Second, this study extends the MO literature by showing the impact of GMO on two types of GI and OP. Third, this study contributes to the MO literature by addressing the mixed relationship between this orientation and performance by measuring OP based on the balanced scorecard approach. Finally, this study sheds light on the positive effects of GMO in logistic industries.

The structure of the study is as follows. Section 2 provides theoretical concepts and the hypothesis development. Section 3 reveals the methodology used. Section 4 demonstrates the results. Then, the discussion Section 5 follows. Section 6 concludes the article and provides avenues for future study.

## 2. Theoretical Background and Hypothesis Development

*2.1. Underlying Theories and Theoretical Concepts*

2.1.1. Green Market Orientation

GMO is a sustainable form of MO. It is a marketing philosophy that is examined intensively in the marketing literature. According to Narver and Slater [20], MO is conceptualized as an element of organizational culture that permits organizations to effectively and efficiently exhibit behaviors that create value for customers and, as such, achieve superior performance. In this conceptualization, MO consists of three behaviors: customer orientation, competitor orientation, and inter-functional coordination [38]. Customer orientation is the belief in prioritizing customers' interests [39], which results in a deep understanding of customers' needs and demands. Consequently, the organization achieves superior performance [20]. Competitor orientation is understanding competitors' strengths, weaknesses, abilities, strategies, and responsive activities [40]. Inter-functional coordination is the integration and collaboration between various departments within organizations through interaction and communication [20,41]. In the literature, MO is conceptualized as a second-order construct, which consists of three separated dimensions: customer orientation, competitor orientation, and inter-functional coordination [42–44].

Borazon et al. [19] draw upon the definition of Narver and Slater [20] to conceptualize GMO as an element of organizational culture, consisting of three separate behavioral components: green customer orientation, green competitor orientation, and green inter-functional coordination. Wang [18] provided the definitions of the three concepts as follows. Green customer orientation refers to behaviors that address environmental change, which allows the organization to satisfy the environmental protection concerns of customers. Competitor orientation is the behavior relating to the acquirement, dissemination, and processing of competitors' environmental strategies and actions, which results in a given company's ability to outperform its competitors. Green inter-functional coordination refers to the collaboration of different departments within the organization to generate, collect, and disseminate market intelligence relating to environmental issues.

2.1.2. Exploitation and Exploratory Green Innovation

Innovation refers to an organization's degree of novelty with respect to ideas, practices, and objects [45]. Innovation can entail the modifications of current products/services and the discovery of new products/services that are valuable and meet customers' demands [46,47]. In the innovation-related literature, the ambidextrous innovation view has gained much attention. It refers to the organizational behaviors that simultaneously improve exploratory and exploitation-based innovation [48]. In addition, according to this view, innovation

varies based on the degree of novelty, e.g., exploratory and exploitation innovation [49,50]. Exploratory innovation refers to the degree of knowledge discovery, which aims to develop new products and services serving for the emerging demands of customers. In contrast, exploitation innovation refers to modifying current products/services to serve existing customers based on existing knowledge of the organization [51]. In the innovation literature, exploratory innovation can refer to radical innovation, while exploitation innovation is incremental innovation [52,53].

GI is a specific form of innovation that focuses on environmental aspects. It refers to environmental innovation or eco-innovation. There are several approaches that have been used to define this concept. One of the most popular definitions, according to Cuerva et al. [54], is that GI allows organizations to introduce new ideas, products, and processes, which reduce and avoid negative impacts on the ecological environment, thereby allowing the organization to achieve their sustainability goals [55,56]. Next, GI refers to the process contributing to the creation of new productions and technologies to reduce the environmental risks and negative consequences caused by resource exploitation [57]. Lastly, GI is defined as the development of environmentally friendly products and processes [58] through the adoption of environmental management practices aiming to reduce emissions and the consumption of resources (e.g., electricity, water, and raw materials) [59].

Similar to innovation, the ambidextrous view argues that GI is classified into two separate approaches: exploratory GI and exploitational GI [26]. Zhao et al. [27] defined the two concepts as follows. Exploratory GI is the modification of current technology or products in order to improve the green aspects of organizations and allow the organization's activities to be more environmentally friendly. Exploitational GI refers to introducing new technology to protect the environment and improving green aspects of organizations.

Innovation can be found in two forms, namely, closed and open approaches. An open approach to innovation suggests that innovation can emerge from outside of the organization [60]. Open innovation is the use of both internal and external resources to increase the level of innovation, whereas commercialization is the application of the results of innovation to the market to generate economic performance [61]. Open innovation is the process of accelerating internal innovation by the use of external knowledge sources [61]. Therefore, this study relies on the open innovation literature to argue that GMO fosters green market knowledge. This knowledge stems from outside the organization; particularly, it originates from the customers and competitors. Thanks to this knowledge, exploratory and exploitational GI emerge, resulting in superior performance.

### 2.1.3. Organizational Performance

OP is the most frequent dependent variable in management research. Due to its popularity, according to Richard et al. [62], researchers usually ignore the provision of its definition in their studies. Several approaches have been employed in an attempt to define this variable. First, Nazarian et al. [63] refer to OP as the measure of achieving organizational objectives. Second, Lee et al. [64] consider this variable as the outcome with which organizations attempt to use relevant strategies and techniques to achieve their goals. Third, Al-Weshah et al. [65] define it as the degree to which organizations achieve their objectives. Fourth, Hussein et al. [66] considered it to correspond to the outcomes of various organizational processes implemented in daily operations. In addition, OP can refer to the outputs that organizations' effectively manage and their ability to deliver value to their customers and stakeholders [67]. Lastly, it is considered to represent the outputs relating to the effectiveness and efficiency of organizations [68].

Researchers in management fields consider OP as an indicator of organizational success [69]. As a result, it is crucial to evaluate OP accurately [70]. In the literature, several approaches have been used to evaluate this performance metric. In the early studies, OP was evaluated based on financial ratios such as the return on investment, return on assets, return on stocks, and earnings per share [71,72]. However, this mode of evaluation is criticized for its short-term orientation and incomprehensiveness [73]. Moreover, the evaluation

of performance based on financial aspects emphasizes past performance and, as such, limits the evaluation of future growth [74]. In this regard, in addition to financial aspects, it is suggested that non-financial aspects should be incorporated into OP [75]. The balanced scorecard approach developed by Kaplan and Norton [76] is an innovative and radical approach that aims to measure OP comprehensively. In this approach, four performance aspects of an organization are considered: financial, customer, business process-related, and learning and growth perspectives [77]. The first perspective focuses on the measurement of financial performance, such as the organization's economic added value, return on assets, and return on investment [78]. It provides information on how shareholders perceive the organization [76]. The second perspective emphasizes measuring customer satisfaction [79]. It provides insight into how customers perceive the organization [76]. The third perspective provides performance information relating to the efficiency and effectiveness of organizational processes, which permits the organization to create and enhance business values [80]. The last one reveals information that enables the monitoring of organizational progress (e.g., the sustainability of the ability to change and grow).

Due to its superiority in terms of the evaluation of OP, recently, management scholars have attempted to incorporate this approach to measure OP in their studies (see [81–85]). In line with prior studies, this study measures OP based on balanced scorecard approaches.

### 2.1.4. Natural Resource-Based View

The NRBV is an extension of RBV. In order to gain an insight into the NRBV, it is necessary to examine the RBV. According to the RBV, the differences in the degree of an organization's performance result from the heterogeneity of organizational resources [86]. In addition, according to this view, organizations can be viewed as a pool of tangible and intangible resources that are necessary for creating competitive advantages and superior performance [87]. Organizational resources allow the organization to develop its capabilities, which are complex, tactic, rare, and difficult to imitate. As a result, the organization can take advantage of these capabilities to improve its performance through the generation of differentiation and cost–leadership advantages [88].

From the RBV, Hart [89] further proposed the NRBV, which argues that the development of competitive advantages depends much more on how the organization successfully manages its relationship with the natural environment. It is argued that specific organizational capabilities are essential for their survival. As a result, organizations should constantly search for and renew their capabilities regarding a successful response to the changing business environment [90]. The NRBV suggests that when organizations continuously renew and search for new capabilities to find innovative environmental solutions, they develop their ability to deal with the increasingly stringent constraints imposed by the natural environment [91]. These abilities constitute the improvement of valuable, rare, and imperfectly inimitable organizational capabilities, leading to highly competitive advantages and superior performance outcomes [92]. According to the NRBV, the three competitive advantages that result from proactively orienting towards environmental behaviors are cost reduction, competitor preemption, and future positions [89].

In the era of environmental concerns, organizations are under pressure to align their strategies with social standards to achieve competitive goals [93]. GMO is considered a strategic approach that grants access to increased green market knowledge and allows departments within the organization to work together to exploit this knowledge to provide green solutions that meet customers' demands and enhance their ability to gain insight into competitors' behaviors with respect to offering green logistic solutions. As a result, GMO is argued to improve green market knowledge, which is valuable, rare, and difficult for competitors to imitate. Zhang et al. [94] argued that organizations that successfully apply green knowledge in developing their products/services to address environmental issues find performance implications. The performance implications are discussed based on the NRBV as follows. It is believed that GI permits three types of competitive advantages: cost reduction, competitor preemption, and future positions. First, in the knowledge manage-

ment literature, green knowledge plays a crucial role in fostering two types of GI [95]. GI was revealed to positively affect cost leadership and differentiation advantages [96]. Cost leadership advantages emphasize cost reduction, and differentiation advantages promote first-mover advantages. They both permit organizations to preempt their competitors [97]. Moreover, as argued by Hart [90], organizations gain future positions by aiming to develop novel products and technologies, and as such, they surpass their competitors. It is argued that GI enables future positions [98]. Therefore, drawing upon the NRBV, this study expects that GMO fosters both types of GI, contributing to the competitive advantages of cost reduction, competitor preemption, and future positions. In this regard, logistics SMEs achieve superior OP.

*2.2. Hypothesis Development*

2.2.1. The Direct Relationship between GMO and OP

Prior studies have examined the direct relationship between GMO and various types of performance. For instance, Tjahjadi et al. [99] found that GMO directly leads to the improvement of business performance of manufacturing micro-enterprises and SMEs. Similarly, Borazon et al. [19] revealed that GMO improves the environmental and economic performance of organizations operating in the Taiwanese electronics industry. Wang [18] also demonstrated that GMO increases the green performance of high-tech firms in Taiwan. From these findings, it is clear that GMO positively impacts performance.

In this regard, this study posits that the adoption of GMO permits Vietnamese SMEs to directly improve their OP. GMO leads to the enhancement of green market intelligence obtained from the insight into customer's green needs and competitor's green behaviors, and departments within the organization coordinate to exploit this intelligence to create superior green values for customers [18,19,100]. GMOs lead to green technology adoption to create green values [25]. This technology is believed to improve four aspects of OP as follows. First, as Clark et al. [100] argue, the world nowadays observes increasing green consumerism and mounting pressures from environmental activists. Organizations should demonstrate themselves as green organizations to improve their ability to attract more customers, particularly green customers [101]. The green products manufactured via green technology improve customer satisfaction because such technology satisfies customers' needs regarding organizations' contribution to a social aim (e.g., addressing environmental issues) [102]. Second, green technology reduces the negative impact on the environment and permits energy conservation, thereby leading to cost reductions [103]. In this regard, green technology improves internal processes through cost reductions. Third, pioneering green technology allows first-mover advantages [104], which translates to high economic performance [105]. Lastly, green technology has been suggested to be able to support the growth of an organization. The authors of [106] used a dataset of 5498 manufacturing firms in Italy between 2000 and 2008 and found that, in comparison to firms adopting non-green technology, firms adopting green technology have a higher degree of firm growth. This study argues that GMO allows SMEs to adopt green technology, resulting in the improvement of four aspects of OP. Hence, the first hypothesis is proposed as follows.

**Hypothesis 1 (H1).** *GMO is directly and positively associated with OP.*

2.2.2. The Direct Relationship between GMO and Exploitational and Exploratory GI

GMO consists of three behavioral components that enhance GI in both its exploitational and exploratory aspects. These three components have previously been revealed to improve GI [18].

The first component is green customer orientation, which enhances organizational sensing capabilities through insights into customer green demands. An insight into green demands allows organizations to modify and create new methods that improve the green values of the products delivered to customers through green product innovation [107]. For instance, green product innovation can be achieved through the use of green packag-

ing [108]. Green packaging is an approach that promotes the usage of environmentally friendly materials [109]. In this regard, SMEs can take advantage of this insight to modify their current product packaging by using green-packaging approaches, leading to GI exploitation. In addition, the MO literature suggests that customer orientation is a crucial determinant of exploratory innovation [110]. In this regard, insights from green customer orientation also assist SMEs in improving their ability to introduce new green products through exploratory GI.

The sensing of competitors' green behaviors assists in the identification of innovative directions and green technology shifts [111]. The ability to produce green products and adopt green technologies requires different forms of green knowledge [112]. Green knowledge is typically drawn from competitors because they are sources of different types of green knowledge relating to technology and the processes used to manufacture green products [113]. In addition, due to the environmental pressure resulting from strict environmental regulations and concerns, as well as demands for green products, organizations must comply with social standards to achieve competitive goals [93]. In this regard, if an organization does not know the orientation of its competitors, it is less likely to gain such knowledge and limits GI from flourishing [18].

On the one hand, knowing a competitor's orientation allows an organization to use the obtained green knowledge to improve its ability to modify its current technology and products, which creates additional green values. Moreover, it is argued that when organizations gain insight into competitors' capabilities, they are more likely to be dissatisfied with those capabilities. As a result, organizations invest in new technology to gain new capabilities, which leads to radical innovation [114]. In this regard, green knowledge also permits SMEs to introduce new green technology and, as such, leads to exploratory GI. In summary, green competitor orientation is expected to lead to exploitational and exploratory GI.

According to de Medeiros et al. [115], inter-functional coordination is one of the most crucial factors determining successful GI. Green inter-functional coordination refers to the collaborations between departments and units within an organization that are formed to generate, collect, and disseminate green market intelligence. In the innovation literature, the shared information and knowledge within an organization contribute to the exploitation of existing knowledge [116]. In this regard, green inter-functional coordination encourages the exploitation of existing green knowledge. Exploitational GI emerges because this knowledge is used to improve current green products and services to meet the current green demands of the market [117]. In addition, inter-functional coordination is also suggested to be the discovery and creation of new knowledge [118]. From this perspective, new green knowledge leads to developing new green products and services, which satisfy the green demands of potential and emerging markets, resulting in exploratory GI [117].

Therefore, this study proposes the following two hypotheses.

**Hypothesis 2 (H2).** *GMO is positively associated with exploitational GI.*

**Hypothesis 3 (H3).** *GMO is positively associated with exploratory GI.*

2.2.3. The Relationship between Exploratory and Exploitational GI and OP

GI promotes the reduction of negative impacts on the environment. According to the NRBV, when organizations proactively renew and search for new methods to achieve innovative solutions for the environment, they can expand and gain new capabilities to improve their competitive advantages [91]. In accordance with the NRBV, GI can be argued to be a source of competitive advantages, which leads to performance improvement. A strong initiative in GI in logistic areas leads to green logistic practices such as green warehousing and green transportation [119,120]. Green warehousing permits minimum energy and maximum space usage, leading to cost reduction and improved efficiency, which results in high performance. Green transportation aims to reduce the rate of carbon dioxide emission through the use of environmentally friendly fuel and route optimizations.

It is suggested to be able to reduce costs and improve efficiency [121,122]. In this regard, it has been revealed that green transportation improves performance [123].

Empirical studies show mixed results regarding the relationship between GI and organizational outcomes in terms of performance. For instance, some indicate the positive effects of GI on performance [124–126], while other studies found that GI does not lead to significantly higher performance [127,128]. In addition, a review by Liao et al. [129] indicated that the cultural background (e.g., western versus eastern) and development (e.g., developed and emerging) of a country under study cause a deviation from the established literature examining the relationship between GI and performance. Thus, due to these two aspects, there is a greater need to study the impact of GI on performance in the context of logistic SMEs operating in Vietnam because of the following two reasons: first, the literature insufficiently indicates the relationship between GI and performance; second, Vietnam is an eastern and emerging country.

Thus, drawing upon the NRBV, this study posits that exploratory and exploitational GI lead to superior OP. Therefore, the next two hypotheses are proposed as follows.

**Hypothesis 4 (H4).** *Exploitational GI is positively associated with OP.*

**Hypothesis 5 (H5).** *Exploratory GI is positively associated with OP.*

2.2.4. The Mediating Effects of Exploratory and Exploitational GI

The MO literature demonstrates that the direct relationship between MO and performance is inclusive [130]. Therefore, MO scholars are concerned that the relationship between MO and performance is not only direct but also indirect [131]. A mediating assessment of the relationship between MO and performance sheds light on the underlying mechanism by which MO directly affects performance.

In the MO literature, it is strongly suggested that innovation acts as a crucial mediator in the relationship between MO and performance [29–36]. Scholars have argued that innovation should be taken into account to foster the beneficial effects of MO on performance. In this regard, many MO scholars attempt to establish the mediating effects of innovation on the relationship between MO and performance in different contexts, such as manufacturers and services.

In line with the MO literature, this study argues that exploitational and exploratory GI mediate the relationship between GMOs and OP. One crucial reason is that although GMO refers to rare, valuable, and inimitable resources permitting performance implications, this orientation also induces GI, which indirectly affects performance. First, GI is sourced from green customer orientation. Green customer orientation provides an insight into the customers' expressed needs as well as their latent green needs. Addressing their expressed green needs leads to the development of solutions to satisfy customers' current green needs. Addressing latent needs permits the enhancement of differentiation advantages. It reduces the probability of being involved in price competition [132], which ensures performance sustainability [133]. Therefore, green customer orientation leads to high-performance outcomes because it induces GI in order to satisfy the expressed and latent green needs of the customers. Second, green competitor orientation enables an insight into competitors' green capabilities and behaviors, and as such, it promotes the ability to preempt competitors' opportunities. A proactive green competitor orientation encourages the organizations' competitive edge over their competitors because these organizations exhibit behaviors aiming to alter the competitive structure and behavior of the market to cause disadvantageous situations for competitors [134]. Innovation enhances the positive effects of competitor orientation to the extent that it fosters an increase in the rate of new products and services' development, which bypasses that of existing products and services [135]. In this regard, green competitor orientation encourages GI because it exploits the market intelligence resulting from this orientation in order to alter the competitive landscape by introducing novel green solutions to bypass the green solutions offered by

competitors. Thus, it results in performance implications. Therefore, GMO has indirect effects on performance through innovation. With these arguments, this study proposes the following final hypotheses.

**Hypothesis 6 (H6).** *Exploitational GI mediates the relationship between GMO and OP.*

**Hypothesis 7 (H7).** *Exploratory GI mediates the relationship between GMO and OP.*

Figure 1 shows the proposed research model of this study, which includes seven hypothesized paths.

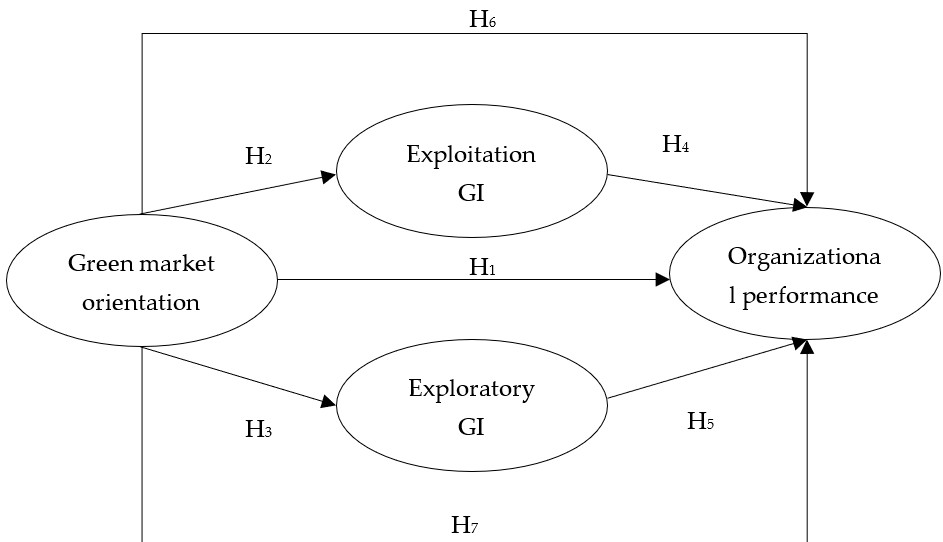

**Figure 1.** Proposed research model.

### 3. Methodology

The study aims to examine the interrelationship between GMO, green exploratory and exploitational innovation, and OP through hypothesis testing. This study employs a quantitative approach to test the proposed hypotheses [136]. In this approach, two sections are considered: the data collection process and data analysis techniques. First, regarding the data collection process, this employed surveys to obtain data. Second, relating to the data analysis methods, this study used structural equation modeling to assess data.

#### 3.1. Data Collection

This study used internet-based approaches. In comparison to conventional survey approaches, this method has some benefits; for instance, (1) its collection time is faster and (2) its method of execution is easier [137]. However, this method is also subject to the weakness of a low response rate [138]. An internet-based survey method is appropriate for this study because target respondents can only be obtained through an email list. According to the General Statistical Office (2020), Vietnam has approximately 38,000 SMEs operating in the logistics industry. These enterprises can be classified as SMEs if they employ less than 50 employees. Due to the lack of financial resources available to conduct a large-scale survey, this study depends on a private agent's database. Specifically, a dataset of 3500 SMEs was selected from a database of 250,000 Vietnamese enterprises. This study focuses on SMEs operating in transportation services, courier express services, and other logistic services (e.g., warehousing services, loading and unloading services, etc.). This data set was acquired via Vietnamese Golden Page [139]. More specifically, this study only acquired the available email addresses of those SMEs.

In March 2022, the collection procedure started. A message was sent to 3500 email addresses. After two months, 391 respondents had finished the survey. Due to missing

data, three observations were omitted. Therefore, 338 observations were employed in the study. The response rate was 9.66% (338/3500).

### 3.2. Data Analysis

This study evaluates the research model using partial least squares structural equation modeling (PLS-SEM). This method seeks to maximize the total variance explained to assess the complex cause-and-effect connections among latent variables [140]. This analysis comprises two modes of evaluation [141]. The examination of the measurement model entails the evaluation of indicator loadings, internal consistency reliability, the convergent validity and discriminant validity of each construct measure, and the discriminant validity of the overall measure [142]. The evaluation of the structural model requires the analysis of collinearity, the model's explanatory capacity, and its predictive accuracy [142].

Zhao et al. [143] proposed an analytical framework to assess the mediating effects. Nitzl [144] and Carrión et al. [145] provided a further approach to examine mediating effects using PLS-SEM. Therefore, this study relies on those mentioned approaches to examine the mediating effects of exploitational and exploratory GI. The PLS-SEM model was evaluated using SmartPLS 3.2.8.

Regarding the appropriateness of sample sizes, PLS-SEM analysis uses various methods to calculate the minimum observation for a sample size. Barclay et al. [146] proposed the ten-time rule. This rule has gained wide acceptance from scholars [147]. This rule specifies that the minimum number of observations should be equal to or greater than ten times the greatest number of arrows pointing to specific latent variables in the measurement or structural model [148]. Based on this criterion, 50 observations are necessary for the analysis if this method is used.

Nonetheless, this rule is questioned because of its inaccurate estimations [147,149]. Therefore, Kock and Hadaya [147] introduced a novel approach, the gamma-exponential method, to determine the minimum sample size using Monte Carlo simulation analysis. According to this method, the minimum number of observations required when the path coefficient is unknown in advance is 146 observations [150]. Moreover, according to Hair et al. [149], the minimal observation of the sample size should be evaluated by statistical methods to detect minimum $R^2$ at a particular significance level. Based on the power table [149], the minimal number of observations required to achieve a statistical power of 80% for identifying at least 0.10 $R^2$ values (with a 5% error probability) with three independent variables in the measurement and structural models is 53. Therefore, the sample size of 338 observations employed in this investigation is sufficient.

### 3.3. Common Method Bias

Podsakoff et al. [151] argued that data collection in the same survey might result in common method bias. Therefore, Harman's single factor test was performed to determine whether this bias was a cause for concern in this study. Common method bias is a concern when a single factor explains the majority of the total variance [152]. The results reveal that 22.40% of the total variance is accounted for by one factor. Therefore, this type of bias is not a cause for concern in this study.

### 3.4. Non-Response Bias

It is a challenge to gain a high response rate in logistics studies [153]. A low response rate increases the concern of non-response bias and, as such, reduces the ability to draw inferences about the population [154]. Non-response bias refers to the situation in which respondents in the sample are different from the non-respondents [155]. When this difference is present, the statistics applied to those respondents do not represent the population, and the results are neither accurate nor valid [156]. Therefore, it is necessary to examine this type of bias before carrying on the main assessment.

According to Clottey and Grawe [157], one of the most used techniques in logistics studies is a comparison between early and late respondents. This study executes a t-test

to compare 25% of the early respondents (e.g., 85 observations at the top of the sample) and 25% of the late respondents (e.g., 85 observations at the bottom of the sample). More specifically, the characteristics such as gender, experience, and the SME's age and size, as well as the respondents' ratings, are included in the text. The results show no significant difference between early and late respondents. Hence, non-response bias was not a cause of major concern in this study.

### 3.5. Measures

This study employed a survey method to collect data; therefore, this study adopted measures from prior studies. Since the target respondents are Vietnamese, the original statements of the measures were translated into Vietnamese. Moreover, the post-translated questionnaire was also examined by two managers with a great deal of experience in the logistics industry before sending it out to the target respondents. The 5-point Likert scale was applied to all the following measures, which range from 1 (strongly disagree) to 5 (strongly agree) (See Appendix A).

### 3.5.1. Green Market Orientation

The measure of GMO adoption was adopted from Wang [18]. In his study, GMO has three dimensions: green customer orientation, green competitor orientation, and green inter-functional coordination. The reason for this operationalization is that the literature defines MO as a culture that drives an organization to exhibit three behaviors, namely, customer orientation, competitor orientation, and inter-functional coordination [20,38]. In this regard, it is argued that these three behaviors reflect MO [158]. From this standpoint, the three behaviors are considered three separate dimensions of MO. According to Polites et al. [159], when a construct consists of more than two dimensions, it should be operationalized as a second-order construct. Thus, this study operationalizes GMO as a second-order construct, which consists of three dimensions.

### 3.5.2. Exploitation and Exploratory Green Innovation

Juo and Wang [28] revised two instruments (e.g., exploratory and exploitation innovation) formulated by Jansen et al. [51] in order to measure the degree of exploratory and exploitational GI. The scope of this study covers SMEs operating in logistic industries. Therefore, these two instruments were revised to measure the exploratory and exploitational GI in logistic industries. Therefore, first, this study changed the term "green products" and/or "green services" to "green logistic services". Second, this study also reduces the ambiguousness of the concept of green logistic services by indicating some practices in green logistic services before asking a given question. These practices drew upon the framework of Jazairy et al. [160].

### 3.5.3. Organizational Performance

The literature suggests that OP is a multidimensional construct [161,162]. However, there is a debate regarding the number of OP's dimensions, and prior studies have failed to systematically determine the validity of the indicators measuring those dimensions [162]. In addition, a construct consisting of two or more dimensions should be operationalized as a second-order construct [159]. Hence, the OP construct should be operationalized as a second-order construct, and this operationalization should draw upon a well-established framework.

Mehralian et al. [81] and Mehralian et al. [8] stated that the construct of OP based on balanced scorecard approaches is a second-order construct including four dimensions: finance, customers, internal processes, and learning and growth. Therefore, in this work, similar to the two mentioned studies, OP was viewed as a second-order construct consisting of the four mentioned dimensions. The measures of four dimensions were drawn from Shahin et al. [163]. They developed these measures by relying on the balanced scorecard approach originally proposed by Kaplan and Norton [76].

### 3.5.4. Control Variables

According to Zhao et al. [164], an organization's age and size affect its performance. Consequently, this analysis includes two variables measuring SMEs' age (AGE) and size (SIZE) in the research model to control for OP. The age of SMEs was measured by the year since it was founded. The number of employees was used to measure the size of SMEs.

## 4. Results

### 4.1. Descriptive Statistics and Respondents' Characteristics

Table 1 indicates the characteristics of the respondents. Male respondents account for the majority of the total respondents. Most respondents have 5 to 15 years of experience in the logistics industry. Most of the logistic enterprises in the sample are located in the South of Vietnam. A total of 42.31% of the total enterprises were founded 10–15 years ago. A total of 49.70% of the enterprises in the sample employ 50–99 employees.

**Table 1.** Respondent's characteristics.

| Characteristic | | Frequency | Percentage |
|---|---|---|---|
| Gender | | | |
| | Male | 248 | 73.37 |
| | Female | 90 | 26.63 |
| Experience | | | |
| | less than 5 years | 21 | 6.21 |
| | 5–15 years | 199 | 58.88 |
| | 15–25 years | 104 | 30.77 |
| | 25–35 years | 13 | 3.85 |
| | more than 35 years | 1 | 0.29 |
| Location | | | |
| | North | 105 | 31.07 |
| | Middle | 11 | 3.25 |
| | South | 222 | 65.68 |
| SME's age | | | |
| | less than 5 years | 12 | 3.55 |
| | 5–10 years | 121 | 35.80 |
| | 10–15 years | 143 | 42.31 |
| | 15–20 years | 43 | 12.72 |
| | 20–25 years | 16 | 4.73 |
| | more than 25 years | 3 | 0.89 |
| SME's size | | | |
| | 10–49 employees | 16 | 4.73 |
| | 50–99 employees | 168 | 49.71 |
| | 100–149 employees | 114 | 33.73 |
| | 150–199 employees | 40 | 11.83 |

Table 2 displays the descriptive statistics applied to the dimenions of the main variables and controlled variables. In this table, the minimum and maximum scores and the mean and standard deviation are shown. In addition, according to this table, the average age of the logistic SMEs is 10.96 years. The average number of employees hired by these enterprises is 99.20.

**Table 2.** Descriptive statistic.

| Dimension | Minimum | Maximum | Mean | Standard Deviation |
|---|---|---|---|---|
| G_CUS_O | 1.000 | 5.000 | 3.193 | 0.746 |
| G_COM_O | 1.000 | 5.000 | 3.240 | 0.755 |
| G_IF_C | 1.250 | 5.000 | 3.696 | 0.796 |
| EXPLT_GI | 1.000 | 5.000 | 3.464 | 0.686 |
| EXPLR_GI | 1.400 | 5.000 | 3.517 | 0.599 |
| OP_FIN | 1.750 | 5.000 | 3.819 | 0.654 |
| OP_CUS | 2.000 | 5.000 | 3.814 | 0.627 |
| OP_IP | 2.000 | 5.000 | 3.812 | 0.614 |
| OP_LG | 2.250 | 5.000 | 3.836 | 0.637 |
| AGE | 3.000 | 28.000 | 10.960 | 4.602 |
| SIZE | 19.000 | 199.000 | 99.200 | 36.356 |

## 4.2. Measurement Model

This study employs a two-stage methodology to evaluate the measurement model [165]. The first step involves assessing the first-order constructions, such as G_COM_O, G_CUS_O, G_IF_C, EXPLR_GI, EXPLT_GI, OP_CUS, OP_FIN, OP_IP, and OP_LG. The second step consists of assessing the second-order constructs, such as GMO and OP.

### 4.2.1. Evaluation of First-Order Constructs

According to Hair et al. [142], an item loading below 0.708 should be removed in further analyses. Table 3 shows that all items are sufficient to be included in the analysis.

**Table 3.** Loadings and cross-loadings of construct items.

|  | G_COM_O | G_CUS_O | G_IF_C | EXPLR_GI | EXPLT_GI | OP_CUS | OP_FIN | OP_IP | OP_LG |
|---|---|---|---|---|---|---|---|---|---|
| G_COM_O_1 | **0.858** | 0.642 | 0.518 | 0.126 | 0.167 | 0.021 | 0.038 | 0.008 | −0.042 |
| G_COM_O_2 | **0.874** | 0.661 | 0.504 | 0.125 | 0.156 | 0.052 | 0.017 | 0.029 | −0.081 |
| G_COM_O_3 | **0.860** | 0.625 | 0.493 | 0.124 | 0.178 | 0.028 | −0.004 | 0.017 | −0.028 |
| G_COM_O_4 | **0.863** | 0.643 | 0.459 | 0.121 | 0.184 | 0.005 | −0.001 | 0.010 | −0.058 |
| G_CUS_O_1 | 0.653 | **0.861** | 0.447 | 0.125 | 0.166 | 0.082 | 0.018 | 0.067 | −0.040 |
| G_CUS_O_2 | 0.629 | **0.879** | 0.485 | 0.180 | 0.114 | 0.093 | 0.077 | 0.064 | 0.039 |
| G_CUS_O_3 | 0.657 | **0.839** | 0.507 | 0.128 | 0.131 | 0.027 | −0.012 | 0.000 | −0.055 |
| G_CUS_O_4 | 0.647 | **0.882** | 0.496 | 0.160 | 0.176 | 0.059 | 0.128 | 0.052 | 0.040 |
| G_IF_C_1 | 0.507 | 0.492 | **0.893** | 0.097 | 0.153 | 0.073 | 0.058 | −0.028 | −0.049 |
| G_IF_C_2 | 0.459 | 0.494 | **0.886** | 0.117 | 0.177 | 0.046 | 0.011 | 0.031 | −0.002 |
| G_IF_C_3 | 0.518 | 0.472 | **0.868** | 0.151 | 0.142 | 0.025 | 0.044 | 0.008 | 0.006 |
| G_IF_C_4 | 0.532 | 0.505 | **0.876** | 0.136 | 0.099 | 0.074 | 0.025 | 0.011 | −0.037 |
| EXPLR_GI_1 | 0.129 | 0.149 | 0.077 | **0.824** | 0.055 | 0.112 | 0.231 | 0.148 | 0.146 |
| EXPLR_GI_2 | 0.108 | 0.125 | 0.141 | **0.795** | 0.087 | 0.090 | 0.238 | 0.109 | 0.163 |
| EXPLR_GI_3 | 0.123 | 0.151 | 0.101 | **0.748** | 0.064 | 0.048 | 0.110 | 0.090 | 0.076 |
| EXPLR_GI_4 | 0.167 | 0.162 | 0.166 | **0.824** | 0.073 | 0.120 | 0.197 | 0.201 | 0.088 |
| EXPLR_GI_5 | 0.023 | 0.101 | 0.065 | **0.774** | 0.115 | 0.046 | 0.162 | 0.166 | 0.083 |
| EXPLT_GI_1 | 0.176 | 0.148 | 0.129 | 0.056 | **0.834** | 0.124 | 0.196 | 0.285 | 0.180 |
| EXPLT_GI_2 | 0.143 | 0.106 | 0.100 | 0.111 | **0.816** | 0.163 | 0.185 | 0.181 | 0.119 |
| EXPLT_GI_3 | 0.174 | 0.150 | 0.146 | 0.098 | **0.831** | 0.132 | 0.197 | 0.173 | 0.176 |
| EXPLT_GI_4 | 0.171 | 0.141 | 0.185 | 0.060 | **0.835** | 0.166 | 0.209 | 0.193 | 0.182 |
| EXPLT_GI_5 | 0.153 | 0.165 | 0.110 | 0.089 | **0.832** | 0.125 | 0.183 | 0.132 | 0.141 |
| OP_CUS_1 | 0.036 | 0.077 | 0.049 | 0.119 | 0.153 | **0.858** | 0.460 | 0.475 | 0.481 |
| OP_CUS_2 | 0.061 | 0.102 | 0.068 | 0.079 | 0.153 | **0.826** | 0.481 | 0.462 | 0.515 |
| OP_CUS_3 | 0.012 | 0.036 | 0.020 | 0.108 | 0.110 | **0.794** | 0.467 | 0.468 | 0.507 |
| OP_CUS_4 | −0.016 | 0.034 | 0.059 | 0.058 | 0.140 | **0.805** | 0.460 | 0.518 | 0.507 |
| OP_FIN_1 | 0.051 | 0.078 | 0.076 | 0.228 | 0.193 | 0.484 | **0.844** | 0.513 | 0.563 |
| OP_FIN_2 | −0.034 | 0.026 | 0.021 | 0.200 | 0.199 | 0.514 | **0.840** | 0.556 | 0.545 |
| OP_FIN_3 | −0.007 | 0.050 | 0.019 | 0.202 | 0.181 | 0.411 | **0.840** | 0.503 | 0.447 |
| OP_FIN_4 | 0.037 | 0.084 | 0.016 | 0.196 | 0.216 | 0.498 | **0.843** | 0.563 | 0.543 |
| OP_IP_1 | 0.037 | 0.042 | 0.034 | 0.209 | 0.163 | 0.517 | 0.521 | **0.802** | 0.483 |
| OP_IP_2 | −0.043 | −0.036 | −0.108 | 0.118 | 0.193 | 0.433 | 0.514 | **0.817** | 0.500 |
| OP_IP_3 | 0.002 | 0.060 | −0.012 | 0.137 | 0.124 | 0.476 | 0.483 | **0.792** | 0.437 |
| OP_IP_4 | 0.055 | 0.111 | 0.089 | 0.133 | 0.265 | 0.470 | 0.534 | **0.830** | 0.504 |
| OP_LG_1 | −0.044 | −0.003 | −0.018 | 0.122 | 0.220 | 0.552 | 0.586 | 0.511 | **0.848** |
| OP_LG_2 | −0.056 | −0.012 | −0.025 | 0.134 | 0.123 | 0.467 | 0.495 | 0.476 | **0.820** |
| OP_LG_3 | −0.061 | 0.024 | 0.001 | 0.124 | 0.166 | 0.496 | 0.494 | 0.474 | **0.828** |
| OP_LG_4 | −0.038 | −0.005 | −0.038 | 0.097 | 0.113 | 0.500 | 0.481 | 0.529 | **0.829** |

The constructs' internal consistency reliability was evaluated using Cronbach's Alpha and composite reliability. All of these numbers exceed the suggested minimum of 0.7. [140]. Table 4 indicates that the constructs' dependability has been established.

**Table 4.** Cronbach's Alpha, composite reliability, and AVE.

|  |  | Cronbach's Alpha | Composite Reliability | AVE |
|---|---|---|---|---|
| First-order constructs | G_COM_O | 0.887 | 0.922 | 0.747 |
|  | G_CUS_O | 0.889 | 0.923 | 0.749 |
|  | G_IF_C | 0.904 | 0.933 | 0.776 |
|  | EXPLR_GI | 0.855 | 0.895 | 0.630 |
|  | EXPLT_GI | 0.887 | 0.917 | 0.689 |
|  | OP_CUS | 0.839 | 0.892 | 0.674 |
|  | OP_FIN | 0.863 | 0.907 | 0.708 |
|  | OP_IP | 0.827 | 0.885 | 0.657 |
|  | OP_LG | 0.853 | 0.899 | 0.691 |
| Second-order constructs | GMO | 0.832 | 0.900 | 0.750 |
|  | OP | 0.858 | 0.903 | 0.699 |

This study examines the convergent validity using average variance extracted (AVE). An AVE value above the 0.5 criteria indicates the satisfaction of the requirement (Hair et al., 2019). Table 4 displays the validation of this requirement.

The discriminant validity of the constructs was determined by assessing the correlation's heterotrait–monotrait (HTMT) ratio. This value must be less than 0.850 for validity to be established. According to Table 5, each of these numbers is less than 0.850. Therefore, the degree of discriminant validity is sufficient.

**Table 5.** HTMT ratios of the correlation between the first-order constructs.

| | G_COM_O | G_CUS_O | G_IF_C | EXPLR_GI | EXPLT_GI | OP_CUS | OP_FIN | OP_IP | OP_LG |
|---|---|---|---|---|---|---|---|---|---|
| G_COM_O | | | | | | | | | |
| G_CUS_O | 0.840 | | | | | | | | |
| G_IF_C | 0.639 | 0.623 | | | | | | | |
| EXPLR_GI | 0.159 | 0.195 | 0.157 | | | | | | |
| EXPLT_GI | 0.222 | 0.190 | 0.178 | 0.116 | | | | | |
| OP_CUS | 0.050 | 0.085 | 0.071 | 0.125 | 0.196 | | | | |
| OP_FIN | 0.052 | 0.092 | 0.051 | 0.274 | 0.267 | 0.667 | | | |
| OP_IP | 0.053 | 0.092 | 0.090 | 0.214 | 0.262 | 0.703 | 0.747 | | |
| OP_LG | 0.071 | 0.056 | 0.043 | 0.162 | 0.211 | 0.718 | 0.718 | 0.707 | |

### 4.2.2. Evaluation of Second-Order Constructs

During this phase, GMO and OP were evaluated. To perform this evaluation, SmartPLS was used to retrieve the latent variable scores of GMO's first-order constructs (G_COM_O, G_CUS_O, and G_IF_C) and OP's first-order constructs (OP_CUS, OP_FIN, OP_IP, and OP_LG) from the first stage. Then, GMO and OP were evaluated using those corresponding constructs as indicators. The evaluation of GMO and OP is comparable to the initial step. The loadings of the indicators with respect to their corresponding constructs are more than the threshold value of 0.708, which suggests that the indicator's validity has been established. In addition, the internal consistency reliability, convergent validity, and discriminant validity of the GMO and OP constructs are well established according to Tables 4 and 6. Consequently, the validity of the GMO and OP constructs are adequate.

**Table 6.** HTMT ratios of the correlation between the second-order and first-order constructs.

| | GMO | EXPLR_GI | EXPLT_GI | OP |
|---|---|---|---|---|
| GMO | | | | |
| EXPLR_GI | 0.205 | | | |
| EXPLT_GI | 0.237 | 0.116 | | |
| OP | 0.063 | 0.230 | 0.284 | |

### 4.3. Structural Model

The structural model was evaluated using bootstrapping with 5000 replacements [140]. First, collinearity issues and predictive accuracy were evaluated. The VIF was assessed to determine the collinearity issue. When this value is below the threshold value of 3, collinearity is not an issue [142]. The predictive accuracy is determined when $Q^2$ is greater than zero [142]. Table 7 indicates that the values of VIF and $Q^2$ satisfy the requirements.

**Table 7.** $Q^2$, $R^2$, and VIF.

| | $R^2$ | $R^2$ Adjusted | $Q^2$ | VIFs | | | |
|---|---|---|---|---|---|---|---|
| | | | | GMO | EXPLR_GI | EXPLT_GI | OP |
| GMO | | | | | 1.000 | 1.000 | 1.073 |
| EXPLR_GI | 0.032 | 0.029 | 0.018 | | | | 1.043 |
| EXPLT_GI | 0.042 | 0.039 | 0.027 | | | | 1.051 |
| OP | 0.104 | 0.090 | 0.063 | | | | |

Second, this study evaluates the strength and magnitude of the hypothesized paths. Figure 2 demonstrates that GMO has significant positive correlations with EXPLT_GI (β = 0.206, *p* = 0.001) and EXPLR_GI (β = 0.179, *p* = 0.005). Both EXPLT_GI (β = 0.247, *p* < 0.001) and EXPLR_GI (=0.199, *p* < 0.001) have significant positive relationships with OP. However, the association between GMO and OP is statistically insignificant (β = −0.053, *p* = 0.333). Consequently, Table 8 shows that the data support hypotheses H2, H3, H4, and H5. However, they do not support hypothesis H1.

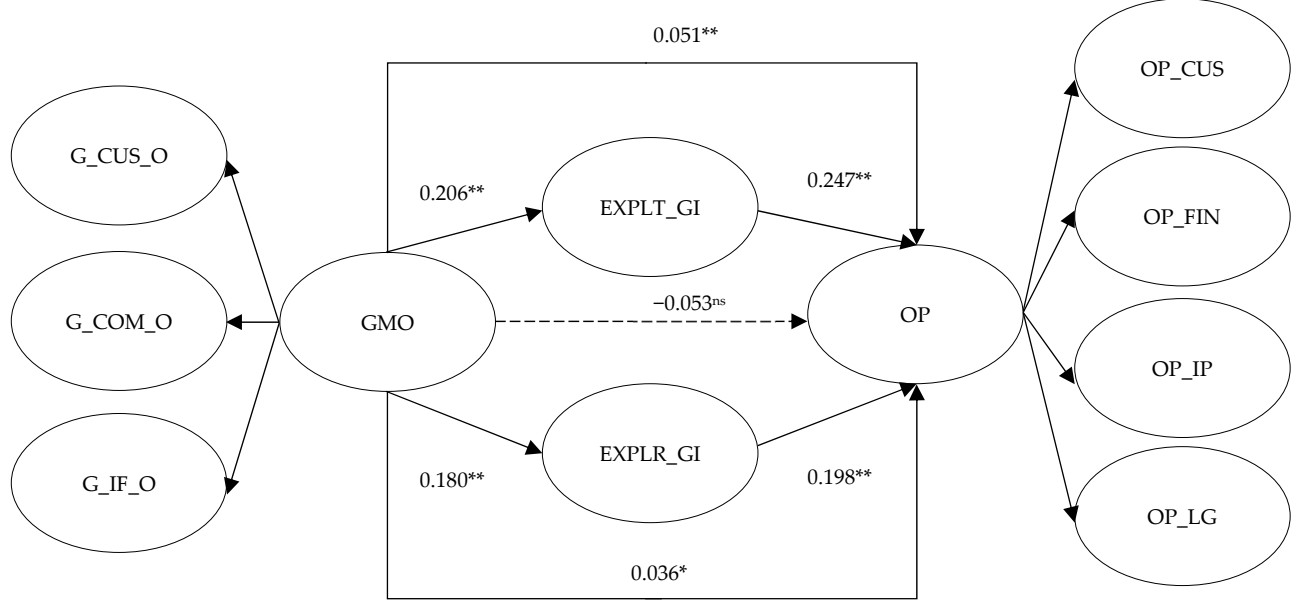

**Figure 2.** Structural model. Note: ** is significant at 0.01. * is significant at 0.05. $^{ns}$ is insignificant.

**Table 8.** Hypothesis testing's results.

| Hypotheses | Paths | Coefficient | t-Value | *p*-Value | Lower Limit | Upper Limit | Results |
|---|---|---|---|---|---|---|---|
| H1 | GMO -> OP | −0.053 | 0.968 | 0.333 | −0.159 | 0.055 | Rejected |
| H2 | GMO -> EXPLT_GI | 0.206 | 3.192 | 0.001 | 0.079 | 0.333 | Accepted |
| H3 | GMO -> EXPLR_GI | 0.180 | 2.791 | 0.005 | 0.054 | 0.306 | Accepted |
| H4 | EXPLT_GI -> OP | 0.247 | 4.730 | 0.000 | 0.144 | 0.348 | Accepted |
| H5 | EXPLR_GI -> OP | 0.198 | 3.861 | 0.000 | 0.099 | 0.304 | Accepted |
| H6 | GMO -> EXPLT_GI -> OP | 0.051 | 2.875 | 0.004 | 0.020 | 0.088 | Accepted |
| H7 | GMO -> EXPLR_GI -> OP | 0.036 | 2.134 | 0.033 | 0.009 | 0.074 | Accepted |
| Control | AGE -> OP | −0.008 | 0.142 | 0.887 | −0.118 | 0.098 | - |
| Control | SIZE -> OP | −0.011 | 0.202 | 0.840 | −0.124 | 0.096 | - |

The outcomes also reveal that the intervening effects of EXPLT_GI (β = 0.051, *p* = 0.004) and EXPLR_GI (β = 0.036, *p* = 0.033) on the relationship between GMO and OP are statistically significant. In addition, their confidence intervals exclude zero (e.g., [0.020; 0.088] and [0.009; 0.074]). Lastly, the direct association between GMO and OP is insignificant when two types of GI are added to control for this path. Thus, according to Table 8, EXPLT_GI and EXPLR_GI fully mediate the link between MARO and OP. Therefore, the data support hypotheses H7 and H8.

## 5. Discussions

### 5.1. Theoretical Implications

This study aims to draw upon the NBRV to shed light on the impact of GMO on OP through two types of GI—exploitational and exploratory GI—in the context of logistics SMEs operating in Vietnam. The results indicate that all the data except the first set support

all the proposed hypotheses. The first hypothesis proposes a direct and positive relationship between GMP and OP. However, the analysis does not support this hypothesis. This finding contradicts prior studies [19,100], which indicate the direct relationship between GMO and business performance and GMO and economic performance. One reasonable explanation is the uniqueness of the logistics providers under this study's scope. It was indicated that customers are less likely to pay extra for green solutions in logistic services [166]. In this regard, when selecting logistic services, the price of these services is more likely to outweigh the environmental benefits of green logistics. In addition, a recent study indicated that the situation has not changed significantly [167]. It suggests that currently, logistic SMEs are less likely to receive direct benefits by adopting GMO because the customers consider the most important factor for selecting the logistic provider to be the price rather than the environmental benefits. Therefore, the logistic SMEs adopting GMO are less competitive with those enterprises that do not adopt this orientation because adopting environmental management practices incurs extra costs. Therefore, this finding implies that logistic SMEs adopting GMO cannot directly achieve performance benefits.

The second and third hypotheses state that GMO adoption is positively related to exploitational and exploratory GI. The results from the analyzed data support these hypotheses. They are in line with prior studies [18,19,25], which indicated a positive relationship between GMO and GI. These results imply that logistic SMEs in Vietnam achieve improvements in both exploitational and exploratory GI due to their adoption of GMO. Adopting GMO results in high green market intelligence. This intelligence is beneficial to the GI of logistic SMEs in two ways. First, it modifies the current green approaches in their logistic services, which results in less damage to the environment. Second, it introduces novel green approaches in SMEs' logistic services, and as a result, it eliminates and/or reduces the negative impact of their services on the environment.

These results somewhat reflect the current condition in Vietnam. For instance, logistics providers shift their mode of transportation from road to water transport because this reduces greenhouse gas emissions. In addition, some providers power their warehouses by installing solar panels. Finally, some providers attempt to use vehicles fueled by electricity.

The fourth and fifth hypotheses anticipated a positive relationship between exploitational and exploratory GI and OP. The findings support those hypotheses. They are consistent with prior study results [125–127], which reveal the beneficial effects of GI on performance. These findings provide the following practical implications. Logistics SMEs acquire OP-related benefits thanks to exploitational and exploratory GI. Exploitational and exploratory GI allow logistic SMEs to modify their current and introduce novel green approaches in their logistic services, such as modifications and the introduction of green transportation and green warehousing. Green transportation permits the reduction of carbon dioxide emissions, which in turn results in cost reduction and efficiency improvement. Green warehousing aims to minimize the energy consumed and maximize the space used, which leads to cost reduction and improved efficiency. They are both crucial determinants of logistic SMEs' performance.

In Vietnam, green transportation can be referred to as the adjustment of the mode of transportation. Vietnam has a very long coastline—approximately 3260 km. Therefore, switching from road transport to water transport is a convenient method to reduce logistics costs. In addition, green transportation also refers to the use of electric vehicles for transportation. It is indicated that electric vehicles are more efficient than conventional vehicles (e.g., diesel and gasoline vehicles). Thus, the use of electric vehicles improves the efficiency of logistics transportation. Moreover, in Vietnam, using solar panels to power warehouses can be considered an example of green warehousing. This use has a crucial impact on logistics providers in terms of cost. Vietnam is close to the equator, so the number of sunny days in a year is high. This location is ideal for using solar panels to generate electricity. The use of solar power permits logistics providers to reduce their costs resulting from electricity bills.

The last two hypotheses propose the mediating effects of exploitational and exploratory GI on the relationship between GMO and OP. The findings confirm these two hypotheses. These two types of GI fully mediate the relationship between GMO and OP. They are in line with the findings of Tjahjadi et al. [99], which indicated that GI plays a crucial intervening mechanism in the relationship between GMO and business performance. However, these studies do not distinguish between two GI types based on the degree of novelty. Therefore, this study's results extend this research and have the following crucial practical implications for logistic SMEs. SMEs operating in emerging countries are under the pressure of uncertainty and, therefore, enhancing their survival is crucial [168]. These results seem to address this aspect because they show that GMO has an indirect effect on OP through exploitational and exploratory GI. Although GMO adoption allows logistic SMEs to have high green market intelligence, this intelligence is insufficient for directly improving OP. Instead, this intelligence indirectly affects OP through exploitational and exploratory GI. Thanks to this intelligence, logistic SMEs modify and introduce novel approaches in their logistics services that meet customers' demands with respect to their expressed and latent green needs. In addition, they also take advantage of this intelligence by bypassing their competitor's green strategies. In this regard, logistic SMEs modify and introduce novel approaches in logistic services, such as the modification and/or introduction of green warehousing and transportation, thereby enabling cost reduction, improved efficiency, and improved performance. Therefore, GMO energizes exploitational and exploratory GI with green market intelligence to achieve a superior OP.

These findings also strongly reflect the open innovation literature. Particularly, the literature suggests that innovation can emerge from outside organizations [60]. Open innovation is the use of both internal and external resources to increase an organization's level of innovation, whereas commercialization is the application of the results of innovation to the market to generate economic performance [61]. Open innovation is the process of accelerating internal innovation by the use of external knowledge sources [61]. The findings somewhat support this literature, because it is shown that GMO fosters green market knowledge. This knowledge originates from outside the organization. Particularly, it stems from the customers and competitors. Thanks to this knowledge, exploratory and exploitational GI emerge, resulting in superior performance. These findings highlight the open approach of innovation.

*5.2. Research Contributions*

Regarding the findings, this study contributes to the literature in the following ways. First of all, the NRBV is a compelling theory that helps to understand how a firm gains competitive advantages when it successfully maintains a healthy relationship with the natural environment. Despite that, this theory has been criticized due to its lack of instructions on how resources can be developed, obtained, and exploited effectively [169]. This study addresses this critique, which is a contribution to the NBRV. This study utilized NRBV to demonstrate that GMO is an intangible resource developed when firms address customers' green demands and try to bypass competitors' green strategies. By employing this resource, firms can effectively improve both their exploitational and exploratory GI, which permits competitive advantages. In addition, these advantages result in superior outcomes. Furthermore, in the GI literature, various theories have been integrated in the research on the drivers and consequences of GI to explain the hypothesized paths. In comparison with other theories, the NBRV seems to be under-adopted as a theoretical lens to examine the drivers and effects of GI [170]. In this regard, this study makes a contribution by utilizing the NRBV to indicate GMO as an antecedent and OP as an outcome of GI.

Second, GMO is a specific form of MO that has gained much attention from MO scholars. In the MO literature, the paradigm of MO innovation performance is the most attractive to scholars [29–36]. Examining its mediating effects offers an insight into the mechanisms that enable MO to enhance performance through innovation [37]. MO scholars further this research line by using the ambidextrous view to explore whether MO fuels various types

of innovation and, as such, has an impact on performance [36]. However, GMO scholars seem to be silent with respect to how the ambidextrous view can be integrated into the framework to explain the impact of various types of GI on the relationship between GMO and performance. This compelling view allows researchers to distinguish GIs based on their degree of novelty, such as exploitational GI and exploratory GI. Therefore, this study extends the GMO literature by borrowing the ambidextrous view to identify that GMO fosters both exploitational and exploratory GI, and as such, they both induce performance.

Third, the literature suggests that there is a mixed relationship between MO and performance. One proposed reason for this concerns the approaches used to measure performance [171]. It is recommended that scholars should employ multiple types of performance to gain insight into the relationship [171,172]. Given is GMO a novel construct, research should examine the impact of GMO on various types of performance. GMO scholars have attempted to use various types of performance when assessing this relationship, such as business performance [99], environmental performance [19,166], and economic performance [19]. Measuring performance based on the balanced scorecard approach is argued to be more comprehensive than the prior approaches. Therefore, this study contributes to the literature by assessing the relationship between GMO and OP based on the balanced scorecard approach.

Fourth, MO scholars have argued that the relationship between MO and performance varies according to the industry [173–176]. MO has certain effects on internal and external processes in each industry, contributing differently to each aspect of organizational effectiveness [177]. An industry-specific study can shed light on the beneficial effects of MO on performance in this industry. Given that GMO is a novel concept, there is a lack of understanding of this concept in logistic industries. In this regard, this study contributes to the literature by exploring the mechanism by which GMO affects OP through exploitational and exploratory GI.

### 6. Conclusions, Limitations, and Future Studies

This study aimed to draw upon the NRBV and the ambidextrous view of GI to examine the impact of GMO on OP through exploitational and exploratory GI. The data were collected by employing a survey. There were 338 observations in the testing sample. The results indicate that exploitational and exploratory GI fully mediate the relationship between GMO and OP. This study provides some practical implications to managers interested in logistic SMEs operating in Vietnam. In addition, this study also addresses several gaps in the existing literature.

However, this study also acknowledges some limitations. Therefore, these limitations create avenues for future studies. First, this study was subject to low response rates. A tailored approach to survey procedures proposed by Dillman et al. [155] aims to increase response rates. In this regard, future research should incorporate this approach into the data collection procedure to address this limitation. The second limitation stems from the use of cross-sectional data. Future studies should collect longitudinal data to address this weakness. The third weakness is the inability to generalize the results to countries other than Vietnam. To overcome this weakness, future studies should replicate this research model with data from other countries. Fourth, the measurement of subjective performance may be a cause for concern. Future studies should use objective data to measure performance. Fifth, this study examines the research model of the internal effects on SMEs' OP when employing GMO. In this regard, future studies should add variables measuring the external aspects of SMEs into the model.

**Funding:** This research received no external funding.

**Institutional Review Board Statement:** Not applicable.

**Informed Consent Statement:** Not applicable.

**Data Availability Statement:** The data will be made available on request from the corresponding author.

**Conflicts of Interest:** The authors declare no conflict of interest.

## Appendix A

| | |
|---|---|
| **GMO** | **Green market orientation** |
| **G_CUS_O** | **Green customer orientation** |
| G_CUS_O_1 | Our company continuously seeks to improve the environmental value for our customers. |
| G_CUS_O_2 | Our company periodically revises environmental activities to ensure that they match with our customers' needs. |
| G_CUS_O_3 | Our company supplies customers with environmental activities so that they can get the best from us. |
| G_CUS_O_4 | Our competitive advantage is based on understanding our customers' concerns for the environment. |
| **G_COM_O** | **Green competitor orientation** |
| G_COM_O_1 | In our company, salespeople share information on competitors' environmental strategies. |
| G_COM_O_2 | Our company responds quickly to competitor's environmental strategies. |
| G_COM_O_3 | In our company, top managers discuss the strengths and weaknesses of competitor's environmental strategies. |
| G_COM_O_4 | In our company, when managers have information about competitors' environmental strategies, they quickly share it with others. |
| **G_IF_C** | **Green inter-functional coordination** |
| G_IF_C_1 | In our company, any environmental regulation information from the market is distributed throughout all departments and levels of the company. |
| G_IF_C_2 | In our company, sharing environmental information with other departments is encouraged. |
| G_IF_C_3 | In our company, all departments are integrated to aim to address customer's concerns for the environment. |
| G_IF_C_4 | In our company, the information relating the environmental strategies of current and potential competitors is widely distributed. |
| **EXPLT_GI** | **Exploitational Green Innovation** |
| EXPLT_GI_1 | Our company frequently adjusts the provision of existing green logistic services. |
| EXPLT_GI_2 | Our company regularly implements small adoptions to existing green logistic services. |
| EXPLT_GI_3 | Our company introduces improved existing green logistic services for our green markets. |
| EXPLT_GI_4 | Our company improves our provision's efficiency of green logistic services. |
| EXPLT_GI_5 | Our company increases economies of scale in existing environmental markets through the refinement of existing green logistic services. |
| **EXPLR_GI** | **Exploratory Green Innovation** |
| EXPLR_GI_1 | Our company accepts demands that go beyond existing green logistic services. |
| EXPLR_GI_2 | Our company invests in new green logistic services. |
| EXPLR_GI_3 | Our company experiments with new green logistic services in our local market. |
| EXPLR_GI_4 | Our company commercializes green logistic services that are completely new to our company. |
| EXPLR_GI_5 | Our company frequently utilizes new opportunities in new environmental markets by introducing new green logistic services. |
| **OP** | **Organizational Performance** |
| **OP_FIN** | **Finance** |
| OP_FIN_1 | Our company has been successful in the efficient and effective use of its investment |
| OP_FIN_2 | Our company has been successful in reducing unnecessary costs and wastage |
| OP_FIN_3 | Our company has a good rate of return |
| OP_FIN_4 | Compared to similar companies, the average productivity rate is better in our companies |
| **OP_CUS** | **Customer** |
| OP_CUS_1 | Our company has succeeded in achieving customer satisfaction |
| OP_CUS_2 | Our company has been successful in identifying customers' demands |
| OP_CUS_3 | Our company has been successful in providing customer service |
| OP_CUS_4 | Our company has been successful in addressing customer complaints |
| **OP_IP** | **Internal process** |
| OP_IP_1 | Our company has been successful in improving the quantity and quality of services |
| OP_IP_2 | Our company has succeeded in implementing internal processes in a timely fashion |
| OP_IP_3 | Our company has been successful in research and development |
| OP_IP_4 | Our company has been successful in its working methods |
| **OP_LG** | **Learning and growth** |
| OP_LG_1 | Our company pays appropriate attention to increase the skills and knowledge of staff |
| OP_LG_2 | Our company pays great attention to increasing employee satisfaction |
| OP_LG_3 | Our company has been successful in developing creative ideas |
| OP_LG_4 | Our company pays great attention to identify the staff development needs |

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
