# Peer review of "The Adoption of Green Market Orientation in Logistic Industries—Empirical Evidence from Vietnamese SMEs"

_2199-8531, doi:10.3390/joitmc8040199_

Round 1

Reviewer 1 Report

please check attachment

Author Response

First of all, I would like to thank the reviewers and editors for providing me a second opportunity to revise my manuscript. The below paragraphs address the point-to-point responses. I hope that this version will meet the requirements.

It is noted that the changes were highlighted in red.

Concern #1: Researchers didn't explain much about research gaps. It will be more interesting if the researcher shows the research gap

Response #1: Thank you so much for pointing out this concern. The introduction was revised. Particularly, the gaps were indicated from line 72-77, 81-84, 89-92 respectively. I hope this revision meets your requirement.

Concern #2: Researchers do not show the newness of this research.

Response #2: Thank you so much for pointing out this concern. The introduction was revised. Particularly, I added the summary of the contributions of this study from line 100-108 in the introduction. I hope this revision meets your requirement.

Concern #3: Did the author do a pilot test? If so, were there any changes in the contents of the questionnaire after the pilot test? If there are, it is better to explain these changes.

Response #3: Thank you so much for pointing out this concern. I did not carry the pilot test. However, there are some minor revisions relating to the questionnaire. As I indicated in line 501-503, I translate the questionnaire from English to Vietnamese, and asked two managers having experience in Vietnamese logistics industry to examine. They help me to make a smooth translation to ensure the translated version to be easy to understand. I hope this clarification meets your requirement.

Concern #4: The findings are a good basis for discussion, but they need more conceptualization to make the contribution of the research more evident. Preferably, the author explains the correlation of statistical results with actual conditions in the logistic industry in Vietnam

Response #4: Thank you so much for pointing out this concern. Following your request, I added two paragraph which connects the findings with the current conditions in Vietnam (paragraph 1: 650-654, paragraph 2: 667-677). I hope this revision meets your requirement.

Reviewer 2 Report

This paper studies the impact of green market orientation on the performance of Vietnamese logistics small and medium enterprises (SMEs) draws upon the natural resource-based view, and explores the mediating effects of exploratory and exploitation green innovation (GI). The sample frame is 338 SMEs operating in the logistics industry, and uses partial least squares structural equation modeling to assess data. The results show that these Vietnamese SMEs indirectly improve operational performance through exploratory and exploitation GI, due to their adoption of green market orientation.

The research questions are interesting, and the contribution fills several important gaps in the literature. However, authors need to do some additional work to make their manuscripts ready for publication. My suggestion is “MAJOR REVISION”.

Comments to the Author

Introduction

#1: The motivation needs further explanation.

The authors say “the relationship between adopting environmental management practices and performance in terms of organizational effectiveness is mixed [18]. It questions the effects of GMO on SMEs’ organizational performance (OP). Therefore, it motivates this study to examine GMO effects on SMEs’ OP.”

But Ahinful and Tauringana (2019) believed that there is a significant positive relationship between environmental management practices (energy, water, and materials) and financial performance, while environmental management practices (waste, emissions, and biodiversity) are not significant. In this paper, the author indicates that the environmental pollution of the logistics industry comes from the greenhouse gas emissions generated by freight transport, which is different from Ahinful and Tauringana (2019). Therefore, the motivation of this paper is not very persuasive and needs to be further explained.

#2: The contribution of this paper needs to be explained in the introduction

Theoretical backgrounds and hypothesis developments

#3: The authors say that the research question is based on the natural resource-based view, so the explanation in this part should be combined with this theory rather than literature accumulation. The manuscript seems to need improvement in this respect

Results

#4: In page14, the authors say “Table 5 indicates that the values of VIF and Q2 satisfy the requirements”

   Is it table 5 or table 7? The authors seem to have made a mistake.

#5: The first hypothesis in this paper is “GMO is positively associated with OP”. However, in the empirical analysis, the authors do not seem to have tested the total effect between GMO and OP.

Discussions

#6: “The first hypothesis proposes a direct relationship between GMP and OP. However, the analysis fails to support this hypothesis.”

The authors need to further explain the reasons for this result.

Conclusions, limitations, and future studies

#7: Are the conclusions of this paper general and representative? As the authors say, there are five limitations in this paper, such as: low response rate, limitations of cross-sectional data, limitations of samples, etc.

Author Response

First of all, I would like to thank the reviewers and editors for providing me a second opportunity to revise my manuscript. The below paragraphs address the point-to-point responses. I hope that this version will meet the requirements.

It is noted that the changes were highlighted in red.

Concern #1: The motivation needs further explanation.

Response #1: Thank you for pointing out the concern. The introduction was revised. I revised the following sentences to emphasize the gap of the literature. You can see them in line 72-77, 81-84, 89-92. I hope these revisions meet your requirement.

Concern #2: The authors say “the relationship between adopting environmental management practices and performance in terms of organizational effectiveness is mixed [18]. It questions the effects of GMO on SMEs’ organizational performance (OP). Therefore, it motivates this study to examine GMO effects on SMEs’ OP.”

But Ahinful and Tauringana (2019) believed that there is a significant positive relationship between environmental management practices (energy, water, and materials) and financial performance, while environmental management practices (waste, emissions, and biodiversity) are not significant. In this paper, the author indicates that the environmental pollution of the logistics industry comes from the greenhouse gas emissions generated by freight transport, which is different from Ahinful and Tauringana (2019). Therefore, the motivation of this paper is not very persuasive and needs to be further explained.

Response #2: Thank you for pointing out the concern. I would make an explanation as follows. In the first version, I mean that the adoption of environmental management practices has a contradicting effect on performance. And I cited those authors because they documented those results in their paper not their results. I did not focus on greenhouse gas emissions generated by freight transport alone. However, because of your concern, I also cited some articles suggesting the reduction of greenhouse gas emissions has contracting impacts on performance. Taken together, I personally think those arguments persuasive. You can see it in line 72-77.  I hope these revisions meet your requirement.

Concern #3: The contribution of this paper needs to be explained in the introduction

Response #3: Thank you for pointing out the concern. In the first version, I addressed the contributions of this study in the discussion section. However, due to your concerns, I added a paragraph summarizing contributions. You can see it from line 100-108. I hope these revisions meet your requirement.

Concern #4: The authors say that the research question is based on the natural resource-based view, so the explanation in this part should be combined with this theory rather than literature accumulation. The manuscript seems to need improvement in this respect

Response #4: Thank you for pointing out the concern. I added a paragraph indicating the integration of NBRV to the research model. You can observe it from 244-265. I hope these revisions meet your requirement.

Concern #5: In page14, the authors say “Table 5 indicates that the values of VIF and Q2 satisfy the requirements”. Is it table 5 or table 7? The authors seem to have made a mistake.

Response #5: Thank you for pointing out the concern. You are totally right. This is my mistake. I revised it in line 600. I hope these revisions meet your requirement.

Concern #6: The first hypothesis in this paper is “GMO is positively associated with OP”. However, in the empirical analysis, the authors do not seem to have tested the total effect between GMO and OP.

Response #6: Thank you for pointing out the concern. You are totally correct. I mean the direct effect not total effect. I made a revision. You can see it from line 298. I hope these revisions meet your requirement.

Concern #7: “The first hypothesis proposes a direct relationship between GMP and OP. However, the analysis fails to support this hypothesis.” The authors need to further explain the reasons for this result.

Response #7: Thank you for pointing out the concern. You are totally correct. I mean the direct effect not total effect. I made a revision. You can see it from line 625-626. I hope these revisions meet your requirement.

Concern #8: Are the conclusions of this paper general and representative? As the authors say, there are five limitations in this paper, such as: low response rate, limitations of cross-sectional data, limitations of samples, etc.

Response #8: Thank you for pointing out the concern. I personally think they are representative and they can be generalized to logistics SMEs in Vietnam because of the following reason. First, low-response rate can cause non-response bias. I added a test to examine this bias (section 3.4 line 482). The result suggests this type of bias does cause a major concern. Second, cross-sectional data has one limitation, which does not take into account the effect of time. Panel data overcomes such issue. However, cross-sectional data does not subject to representative issue.

Finally, relating to the sample, this study focuses logistics SMEs operating in Vietnam. And thus the generalization of this study to other country context is limited.

I hope these revisions meet your requirement.

Concern #9:

Liu, F., Fang, M., Park, K., & Chen, X.(2021) Supply chain finance, performance, and risk: How do SMEs adjust their buyer-supplier relationship for competitiveness?. Journal of Competitiveness, 13(4), 78–95. https://doi.org/10.7441/joc.2021.04.05

Liu, F., & Park, K. (2021). Managing firm risk through supply chain dependence: an SME perspective, Journal of Business & Industrial Marketing, 36(12), 2231–2242. https://doi.org/10.1108/JBIM-05-2019-0229

Xu, J., & Liu, F., & Shang, Y. (2020). R&D investment, ESG performance and green innovation performance: evidence from China, Kybernetes, 50(3), 737-756. https://doi.org/10.1108/K-12-2019-0793

Response #9: Thank you so much for indicating some interesting articles. Due to the scope of the paper, I only cited the second mentioned paper.

Reviewer 3 Report

Abstract: Line 15, “thanks to adopting green market orientation”, what does this mean?

Introduction: In this section, most of the focus is given on Vietnam. It is suggested that please set the study context at the world level and then study the country level. Many technical terms were used e,.g. 200/ED-TTG, as a lay reader how I can understand the overall purpose your article. In intro. much information was there but it is difficult to understand the study motivation or study problems. It is suggested that please rewrite this section and highlight the problem and severity of the problem both at local and international levels.

Methodology: The response rate is 9.66%. Please justify it through literature support.

Data Analysis: Table 1. please recheck the percentage, some values are above 100% like experience 100.01. Please check it carefully. Table 2. Descriptive statistics, please calculate the statistics dimensions-wise, not indicators-wise. Table 8. What are the acceptance or rejection criteria? Please add a column of p-value and Lower limit (LL) and upper limit (UL).

Please justify why you perform analysis through second/higher order. E.g., GMO has three dimensions (customers, competitors, and inter-functional), and the same with performance. In GMO each dimension is unique and has different operationalization. Please justify it.

For mediation analysis, what is/are the types of mediation full or partial? For mediation in PLS-SEM, please follow some good approaches like “Mediation Analyses in Partial Least Squares Structural Equation Modeling: Guidelines and Empirical Examples” (DOI 10.1007/978-3-319-64069-3_8) and “Mediation analysis in partial least squares path modeling Helping researchers discuss more sophisticated models” (DOI 10.1108/IMDS-07-2015-0302).

Author Response

First of all, I would like to thank the reviewers and editors for providing me a second opportunity to revise my manuscript. The below paragraphs address the point-to-point responses. I hope that this version will meet the requirements.

It is noted that the changes were highlighted in red.

Concern #1: Abstract: Line 15, “thanks to adopting green market orientation”, what does this mean?

Response #1: Thank you for pointing out the concern. This issue was revised. The whole sentence was change. You can see it from line 13-14. I hope that this revision meets your requirement.

Concern #2: Introduction: In this section, most of the focus is given on Vietnam. It is suggested that please set the study context at the world level and then study the country level. Many technical terms were used e,.g. 200/ED-TTG, as a lay reader how I can understand the overall purpose your article. In intro. much information was there but it is difficult to understand the study motivation or study problems. It is suggested that please rewrite this section and highlight the problem and severity of the problem both at local and international levels.

Response #2: Thank you for pointing out the concern. Following your suggestion, I added a paragraph describing Vietnam at international levels. Besides, 200/ED-TTG is the name of the decree, which was passed by the government. The motivation of the study was revised too. Particularly, I revised the following sentences to emphasize the gap. You can see them in line 72-77, 81-84, 89-92. I hope these revisions meet your requirement.

Concern #3: The response rate is 9.66%. Please justify it through literature support.

Response #3: Thank you for pointing out the concern. I added a test to address the issue resulted from low response rate. You can see it in section 3.4 (line 482-497). I hope these revisions meet your requirement.

Concern #4: Table 1. please recheck the percentage, some values are above 100% like experience 100.01. Please check it carefully. Table 2. Descriptive statistics, please calculate the statistics dimensions-wise, not indicators-wise. Table 8. What are the acceptance or rejection criteria? Please add a column of p-value and Lower limit (LL) and upper limit (UL).

Response #4: Thank you for pointing out the concern. Table 1 was revised and the revised number is highlighted in red (see 552). The required information was added to Table 8 (see 619). I hope these revisions meet your requirement.

Concern #5: Please justify why you perform analysis through second/higher order. E.g., GMO has three dimensions (customers, competitors, and inter-functional), and the same with performance. In GMO each dimension is unique and has different operationalization. Please justify it.

Response #5: Thank you for pointing out the concern. You can observe the justification in section 3.5.1 and 3.5.3 (see line 509-516 and 527-532). I hope these revisions meet your requirement.

Concern #6: For mediation analysis, what is/are the types of mediation full or partial? For mediation in PLS-SEM, please follow some good approaches like “Mediation Analyses in Partial Least Squares Structural Equation Modeling: Guidelines and Empirical Examples” (DOI 10.1007/978-3-319-64069-3_8) and “Mediation analysis in partial least squares path modeling Helping researchers discuss more sophisticated models” (DOI 10.1108/IMDS-07-2015-0302).

Response #6:  Thank you for pointing out the concern. This mediation is full because there is insignificant relationship of indirect effects. This study relies on Zhao et al (2010) approach. I revised a paragraph (line 452-456) to illustrate this. Besides, I also cited two mentioned articles because they both based on Zhao et al (2010). I hope these revisions meet your requirement

Round 2

Reviewer 2 Report

I would like to thank you for the opportunity to read and review your manuscript, again. In fact, I can see the significant improvement in your manuscript. Finally, I wish your team every success in the research. Thank you, again.